# Test-Time Training on Graphs
# with Large Language Models (LLMs)

## ABSTRACT

Graph Neural Networks have demonstrated great success in various fields of multimedia. However, the distribution shift between the training and test data challenges the effectiveness of GNNs. To mitigate this challenge, Test-Time Training (TTT) has been proposed as a promising approach. Traditional TTT methods require a demanding unsupervised training strategy to capture the information from test to benefit the main task. Inspired by the great annotation ability of Large Language Models (LLMs) on Text-Attributed Graphs (TAGs), we propose to enhance the test-time training on graphs with LLMs as annotators. In this paper, we design a novel Test-Time Training pipeline, LLMTTT , which conducts the test-time adaptation under the annotations by LLMs on a carefully-selected node set. Specifically, LLMTTT introduces a hybrid active node selection strategy that considers not only node diversity and representativeness, but also prediction signals from the pre-trained model. Given annotations from LLMs, a two-stage training strategy is designed to tailor the test-time model with the limited and noisy labels. A theoretical analysis ensures the validity of our method and extensive experiments demonstrate that the proposed LLMTTT can achieve a significant performance improvement compared to existing Out-of-Distribution (OOD) generalization methods.

## CCS CONCEPTS

• **Computing methodologies** → *Artificial intelligence*; *Machine learning*.

## KEYWORDS

Test Time Training, Large Language Models, Graph Neural Networks, OOD Generalization

## 1 INTRODUCTION

Graph is a kind of prevalent multi-modal data, consisting of modalities of both the topological structure and node features [30, 38]. Text-Attributed Graphs (TAGs) are graphs of which node attributes are described from the text modality, such as paper citation graphs containing paper descriptions and social network data including user descriptions. As a successful extension of Deep Neural Networks (DNNs) to graph data, Graph Neural Networks (GNNs) have demonstrated great power in graph representation learning, and have

*ACM MM, 2024, Melbourne, Australia*
© 2024 Copyright held by the owner/author(s). Publication rights licensed to ACM.
ACM ISBN 978-x-xxxx-xxxx-x/YY/MM
https://doi.org/10.1145/nnnnnnn.nnnnnnn

achieved revolutionary progress in various graph-related applications, such as social network analysis [16], recommendation [39, 64] and drug discovery [8, 15]. Despite remarkable achievements, GNNs have shown vulnerability in Out-Of-Distribution (OOD) generalization, as it is observed that GNNs can confront significant performance decline when there exists distribution shift between the training phase and the test phase [19, 33].

Increasing efforts [56, 58] have been made to address the Out-Of-Distribution (OOD) challenge on graphs. A majority of these methods aim at increasing the models' capability and robustness via data augmentation techniques designed based on heuristics and extensive empirical studies [28, 55, 61]. Meanwhile, some researchers have investigated to improve the model's generalization capability via adversarial training strategies [58] and the principle of invariance [56]. Nevertheless, these approaches [56, 58] require interventions during the training phase and can hardly make the continuous adaptability to the real-time data within the constraints of privacy, resources, and efficiency. This gap has prompted the development of Test-Time Training (TTT) [31, 44], which aims to dynamically adapt to continuously presented test data based on an unsupervised learning task during the test phase.

Test-Time Training (TTT) have demonstrated great potential in alleviating OOD generalization problem. Fully Test-Time Training (FTTT) [25, 49] is the extension of TTT. This kind of post-hoc method is more suitable for real-world applications due to its plug-and-play simplicity, which does not interfere with the expensive training process required for pre-trained backbones. Traditional FTTT aims at adapting the pre-trained model to accommodate test data from different domains within an unsupervised setting. However, the design of the unsupervised training phase entails stringent criteria: it must ensure that the unsupervised task complements the main task without causing overfitting to the model and neglecting the main task. Additionally, unsupervised tasks must implicitly capture the distribution of the test data. Devising such an unsupervised training strategy poses a significant challenge. A natural solution is to utilize the same training strategy as the main task in the test phase, i.e., supervised learning. Meanwhile, a recent study [12] has shown that incorporating a limited number of labeled test instances can enhance the performance across test domains with a theoretical guarantee. This motivates us to introduce a small number of labels at test time to further advance the model performance on OOD graphs.

In the FTTT scenario, with continuous arrival of data during testing, human annotation cannot handle this situation flexibly and efficiently. Fortunately, Large Language Models (LLMs) have achieved impressive progresses in various applications [6, 14, 17]. including zero-shot proficiency in annotation on text-attributed graphs [7]. With the assistance of LLMs, only a few crucial nodes are chosen and assigned pseudo labels. Then, FTTT is executed using the same training approach as the main task. This method avoids

the need for intricate unsupervised task designing. Therefore, in this work we propose a novel method to leverage the annotation capability of LLMs to advance test-time training, so as to alleviate the OOD problem on graphs. However, to achieve this goal, we face tremendous challenges: (1) How to select nodes for annotation with LLMs given a limited budget? The studied problem in this paper is different from that in [7]. For node selection, in addition to the importance of the characteristics of LLMs and the test data, the predictions of the pre-trained model on test nodes can also provide crucial signals. (2) How to effectively adapt the pre-trained model under the noisy and limited labels? The labels generated by LLMs are noisy [7]. Therefore, it is essential to design a training strategy which is able to simultaneously utilize a small number of noisy labeled nodes and the remaining unlabeled nodes during test time.

To tackle these challenges, we introduce a Fully Test-Time Training with LLMs pipeline for node classification on graphs, LLMTTT . During the selection of node candidates, different from traditional graph active node selection methods, LLMTTT introduces a hybrid active node selection strategy, which considers node diversity, node representativeness, and the prediction capacity of the pre-trained GNN simultaneously. Meanwhile, to leverage both the noisy labeled nodes and unlabeled nodes, LLMTTT designs a two-stage test-time training strategy. Our main contributions can be summarized as follows:

- We introduce a new pipeline, LLMTTT , from the graph OOD problem. In LLMTTT , we use LLMs as annotators to obtain pseudo labels. These labels are then used to fine-tune the pre-trained GNN model during test time.
- We develop a hybrid active node selection which considers not only the node diversity and representativeness on graphs but also the prediction signals from the pre-trained model.
- We design a two-stage training strategy for the test-time model adaptation under the noisy and limited labeled samples.
- We have conducted extensive experiments and theoretical analysis to demonstrate the effectiveness of LLMTTT on various OOD graphs.

## 2 PRELIMINARY

This section provides definitions and explanations of key notations and concepts in this paper. First, primary notations and the pipeline of traditional fully test-time training are introduced. Next, we illustrate the proposed LLMTTT pipeline for a more comprehensive understanding of our framework.

In this study, we focus on the node classification task, where the goal is to predict the labels of nodes within a graph and we denote the loss function for this task as $L_m(\cdot)$. Given a training node set $D_s = (X_s, Y_s)$ and a test node set $U_{te} = (X_t)$, where $X$ denotes the node samples and $Y$ indicates the corresponding labels.

**Traditional FTTT pipeline.** Assuming that the model for the node classification task has $K$ layers, which can be denoted as $\theta = \{\theta_1, ..., \theta_K\}$. Given the test data $U_{te}$, the parameters of the learned model will be partially (typically the first $k$ layers of the model are fixed) updated by the SSL task during the fully test-time training phase. We can denote the updated part of model as

$(\theta'_{k+1}, ..., \theta'_K)$. In the inference phase, the model $(\theta_1, ..., \theta_k, ..., \theta'_K)$ is used to make predictions for the test data.

**The proposed LLMTTT pipeline.** Traditional FTTT pipeline aims at adapting a pre-trained model for streaming test-time data under unsupervised settings. However, it is not trivial to design such an appropriate and effective unsupervised task, which is supposed to be positively-correlated to the main training task [44]. In order to solve this problem, we introduce a novel pipeline named LLMTTT , which substitutes a semi-supervised task with the assistance of LLMs, for the unsupervised task during the test-time training phase. The proposed pipeline can be formally defined as follows:

Given a model $f(x; \theta)$, initialized with parameters $\theta_s$ obtained by pre-training on train data. We select most valuable samples under a limited budget from test nodes by a carefully designed hybrid node selection method, denoted as $X_{tr} = ActAlg(X_t)$. Then the selected samples are given pseudo labels by LLMs, denoted as $D_{tr} = (X_{tr}, \hat{Y}_{tr})$ where $\hat{Y}_{tr} = LLM_{anno}(X_{tr})$. After obtaining the labeled test nodes, we employ a two-stage training strategy that incorporates both the labeled test nodes $D_{tr}$ and unlabeled test nodes $D_{tre}$. The LLMTTT task aims to optimize the model as:

$$\theta^* := \underset{\theta}{\arg\min} \left( \mathbb{E}_{(x,\hat{y}) \in D_{tr}} \left[ L_C(f(x; \theta), \hat{y}) \right] + \mathbb{E}_{x \in D_{te}} \left[ L_U(f(x; \theta)) \right] \right), \tag{1}$$

$$\text{where} \quad X_{tr} = \begin{cases} \varnothing, & \text{in FTTT} \\ ActAlg(X_t), & \text{in LLMTTT,} \end{cases} \quad \text{s.t.} \quad |X_{tr}| \leq B, \tag{2}$$

$L_C$ is the cross entropy loss, $L_U$ is an unsupervised learning loss, and $B$ is the budget. $D_{te}$ are the unlabeled nodes in the test data $U_{te}$ that have not been labeled. $\hat{y}$ is the pseudo labels given by LLMs.

## 3 METHOD

In this section, we will introduces the novel LLM-based fully test-time training framework ( LLMTTT ) for the graph OOD problem. We first delineate the overall framework and then detail the specific components of LLMTTT .

### 3.1 An Overview of LLMTTT

The LLMTTT pipeline proposed in this paper is illustrated in the Fig. 1 that consists of three parts: pre-training phase, fully test-time training phase, and inference phase as follows:

**Pre-training phase.** The objective of this phase is to acquire a pre-trained classification model with optimized parameters capable of accurately predicting labels for the train data $D_s$. It is worth noting that only the model parameters $\theta$ and the test data $U_{te}$ are required for the subsequent test-time model adaptation. Therefore, LLMTTT is a model-agnostic framework.

**Fully test-time training phase.** The objective of our proposed approach is to utilize the annotation capabilities of LLMs to enhance test-time training to handle the OOD problem on graphs. We encounter several challenges in achieving this goal: (1) How to select the most valuable nodes for annotation using LLMs within a constrained budget? To address this issue, LLMTTT proposes a hybrid active node selection method incorporating both the knowledge from the pre-trained model and the node characteristics. Detailed illustration is provided in Section 3.2. (2) How to obtain high-quality pseudo labels based on LLMs? Given the candidate set of nodes, the

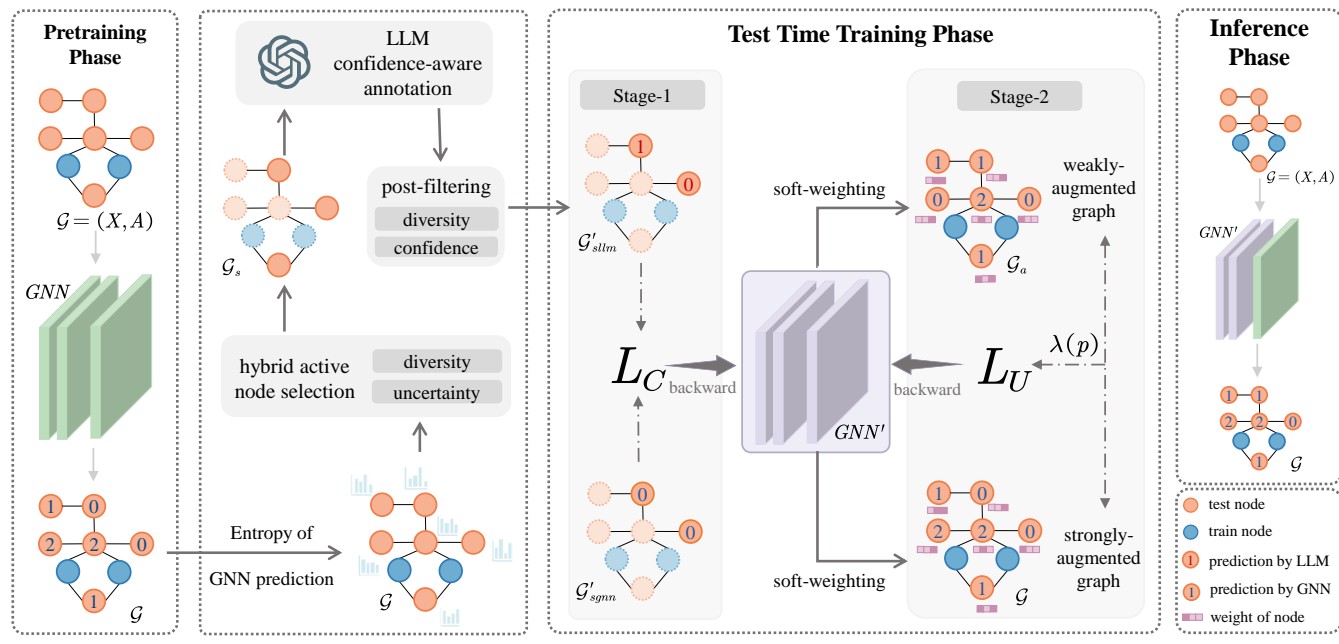

**Figure 1: The overall framework of LLMTTT.**

quality of pseudo labels is crucial. Thus, we enhance the annotation by carefully designing various prompts, as described in Section 3.3. Moreover, the confidence scores from LLMs' predictions are used for further node filtering. (3) How to effectively adapt the pre-trained model under the noisy and limited labels? It is challenging to design a strategy to jointly leverage noisy labels and test samples. To tackle this challenge we propose a two-stage training strategy including training with filtered nodes and self-training with unlabeled data. Additional information is available in Section 3.4. After a two-stage test-time training, the pre-trained model is updated specifically for the test set.

**Inference phase.** During the inference phase, the updated model is utilized to predict the labels for the test data, following the traditional model inference process.

## 3.2 Hybrid Active Node Selection

Node selection is crucial in the design of LLMTTT . To better prompt the model performance under a controllable budget, it is eager to select the most valuable nodes for the test-time training. To achieve this goal, we need to consider not only the characteristics of the test data, but the predictions of the pre-trained model on test data. Thus, LLMTTT proposes a two-step hybrid active node selection method. It consists of uncertainty-based active learning to leverage the important signals from the pre-trained model, and distribution-based active learning that is able to exploit the data characteristics. The details of these two steps are illustrated in the following subsections.

*3.2.1 uncertainty-based active learning.* To fully exploit the potential of model improvement from the test-time model adaptation, LLMTTT targets at nodes that are most difficult for the pre-trained

GNN model to predict. To achieve this, uncertainty-based active learning is designed, which makes use of the prediction uncertainty indicated by prediction entropy [43] to select potential annotation nodes. Unlike other metrics of uncertainty [37, 48], entropy takes into account all class probabilities for a given $x$. Specifically, for each node $v_i$, LLMTTT computes its prediction entropy $Q(v_i)$ based on the prediction results from the pretrained GNN model, and then the nodes with higher prediction entropy are more likely to be selected. The prediction entropy of node $v_i$, $Q(v_i)$ is calculated as follows:

$$Q(v_i) = -\sum_c p(y = c|x_i) \log p(y = c|x_i), \qquad (3)$$

where $c$ denotes the potential labels and $y$ is the predicted label given by GNNs.

*3.2.2 distribution-based active learning.* The nodes selected through uncertainty sampling often exhibit high correlation within the same neighborhood. As a result, the distribution of the selected node set deviates from the original distribution, significantly compromising the diversity and representativeness of the node candidate set [47]. With this rationale, LLMTTT proposes to further refine node selection using distribution-based methods to emphasize the crucial data distribution characteristics. To be specific, a combiantion of PageRank [32] and FeatProp [57] is employed to capture the node distribution from both the structural and feature perspective.

*3.2.3 The Selection Algorithm.* The hybrid active node selection process is summarized in Algorithm 1. In order to select the most valuable $B$ nodes, a scaling factor $\beta$ is introduced to broaden the range of selection in the first step. Initially, LLMTTT filters out $\beta B$ samples where $\beta > 1$, that exhibit the highest level of uncertainty. To consider both structural and feature attributes, we devise a composite active learning score $F(v_i)$ as the criterion for distribution

**Algorithm 1** The Selection Algorithm

**Input:** $X_t, GCN$
**Output:** $X_{tr}$
1: $Y = GCN(X_t)$
2: $Q_{\text{list}} = []$
3: **for** $v_i$ in $X_t$ **do**
4: $\quad Q(v_i) = -\sum_c p(y = c|x_i) \log p(y = c|x_i)$
5: $\quad Q_{\text{list}}.\text{append}(Q(v_i))$
6: **end for**
7: $S_{\beta B} = (X_t)$ for $v_i$ in $\text{sorted}(Q_{\text{list}}, \text{reverse=True})[: \beta B]$
8: $F_{\text{list}} = []$
9: **for** $v_i$ in $S_{\beta B}$ **do**
10: $\quad F(v_i) = Score_{pagerank}(v_i) + \alpha \times Score_{featprop}(v_i)$
11: $\quad F_{\text{list}}.\text{append}(F(v_i))$
12: **end for**
13: $X_{tr} = (X_t)$ for $v_i$ in $\text{sorted}(F_{\text{list}}, \text{reverse=True})[: B]$
14: **return** $X_{tr}$

selection. Subsequently, $B$ samples that exhibit both uncertainty and diversity are selected.

## 3.3 Confidence-aware High-quality Annotation

Given the set of selected nodes, the quality of their pseudo labels plays an important role in the performance after test-time training, based on the empirical study in Section 5.3.1. Therefore, it is imperative to make full use of LLMs and the pretrained GNN to obtain high-quality annotations after acquiring the candidate node set via the hybrid active learning. Inspired by existent exploration of LLMs on graphs [6, 7, 14], LLMTTT proposes to prompt based on the "few-shot" strategy, which is described in Appendix B. Specifically, information of some labelled nodes from the training set serves as a part of the prompt. Moreover, the prediction results from the pretrained GNNs are also included into the prompt. In addition, to evaluate the quality of LLM's annotations, we further request the prediction confidence for the pseudo labels from LLMs.

## 3.4 Two-Stage Training

After LLMs' annotation on the selected nodes, LLMTTT moves to the next phase, test-time training phase. The proposed LLMTTT creatively suggests utilizing pseudo labels for semi-supervised training during test time instead of an unsupervised training strategy. However, given the LLMs' annotation budget, the pseudo labels are too few to effectively adapt the model, which may even lead to a biased adaptation. To tackle this challenge, we propose to further design test-time training by integrating unsupervised learning with supervised learning to better leverage the information from all test nodes. In a nutshell, during test-time training phase, LLMTTT first trains the model with filtered nodes so as to reduce the impact from the noisy labels, and then leverages the self-training that can incorporate the information from the unlabeled data.

*3.4.1 Stage 1: Training with filtered nodes.* The pseudo labels generated by LLMs may be noisy and consequently affect the model. The pseudo labels are not entirely accurate. Therefore, we obtain the confidence of the LLMs' prediction through confidence-aware high-quality annotation in Section 3.3. To mitigate the potential

impact from the noisy pseudo labels, LLMTTT propose to do a node filtering by excluding nodes based on confidence scores. However, it may cause label imbalance in the annotated node set. To avoid this issue, LLMTTT proposes to take the label diversity into consideration during the node filtering process.

To quantify the change in diversity, we adopt the Change of Entropy (COE), inspired by [7]. It measures the shift in entropy of labels when a node is removed from the set. Specifically, assuming that the current set of selected nodes is denoted as $V$, COE can be defined as $COE(v_i) = H\left(\hat{y}_{V-\{v_i\}}\right) - H(\hat{y}_V)$ where $H(\cdot)$ is the Shannon entropy function [43], and $\hat{y}$ denotes the annotations generated by LLMs. A larger COE value indicates that the removal of a node from the node set has a more pronounced impact on the diversity of the node set. To conclude, we integrate COE with the confidence score provided by LLMs to effectively balance both diversity and annotation quality. The final filtering score of each label can be expressed as $Score_{filter}(v_i) = Score_{conf}(v_i) - \gamma \times COE(v_i)$. The annotated nodes with relatively-high filtering score are selected for the few-shot test-time learning. Then we Then the filtered nodes, along with their corresponding pseudo labels, are utilized as supervision for model adaption. In this case, the cross-entropy loss $L_C$ is employed in stage 1.

*3.4.2 Stage 2: self-training with unlabeled nodes.* To alleviate the potential biased model adaptation from the limited noisy labeled annotated by LLMs, the proposed LLMTTT designs an additional self-training stage, which aims at leveraging the information from large amount of unlabeled test data.

Inspired by Softmatch [5], to fully leverage the unlabeled data information, we further perform self-training on the fine-tuned GNN model with the unlabelled test data. Specifically, an augmented view is generated via DropEdge. Next, a weighted cross-entropy loss are computed between the original view and augmented view. Intuitively, the more confident the prediction is, the more important role will this node make in the weighted cross-entropy loss. Formally, we denote the weighted cross-entropy loss $L_u$ as follows:

$$L_u = \sum_{i=1}^{N} \lambda\left(p\left(y \mid x_i\right)\right) H\left(p\left(y \mid x_i^a\right), p\left(y \mid x_i\right)\right) \quad (4)$$

where $p(y|x)$ denotes the model's prediction, $x_i$ is an unlabelled test node in $D_{te}$, $x_i^a$ represents the augmented view and $x_i$ is the original data. $y$ is the prediction given by updated model. $\lambda(p)$ is the sample weighting function where $p$ is the abbreviation of $p(y|x)$. $N$ is the size for unlabeled data.

The uniform weighting function is vital to this process. An ideal $\lambda(p)$ should accurately represent the original distribution while maintaining both high quantity and quality. Despite its importance, $\lambda(p)$ is rarely explicitly or adequately defined in existing methods. Inherently different from previous methods, we assume that the weight function lambda follows a dynamically truncated Gaussian distribution followed by [5]. More detailed is provided in Appx. F.

## 4 THEORETICAL ANALYSIS

Compared to the traditional TTT pipeline, LLMTTT introduces supervision into the model adaptation process. This section theoretically demonstrates that incorporating labelled test samples

provided by LMMs during the test-time training phase can significantly improve the overall performance across the test domain. This also provides a theoretical guarantee for the proposed LLMTTT .

To simplify the theoretical analysis, we consider the main task as a binary classification problem. Given a domain $X$ with two probability distributions $D_1$ and $D_2$, $h : X \rightarrow \{0, 1\}$ is a hypothesis serving as the prediction function from domain $X$ to a binary label space. Let $\mathcal{H}$ denote a hypothesis class with VC-dimension $d$. We employ the $\mathcal{H} \triangle \mathcal{H}$-distance as detailed in [1], offering a fundamental metric to quantify the distribution shift between $D_1$ and $D_2$ over $X$. The discrepancy between $h$ and the true labeling function $g$ under distribution $D$ is formally expressed as $e(h, g) = \mathbb{E}_{x \sim D}[|h(x) - g(x)|]$, commonly known as the domain error $e(h)$.

Building upon two lemmas [12] provided in Appx. G, we establish theoretical bounds under the LLMTTT setting when minimizing the empirical weighted error using the hypothesis $h$. Thm. 1 characterizes the error bounds in the LLMTTT setting, which can be formally expressed to quantify the generalization error. Expanding on this, Thm. 2 establishes the upper bound of the error that can be effectively minimized by integrating a portion of the labeled test data compared with FTTT.

**Theorem 1.** *Considering data domains $X_s$, $X_t$, let $S_i$ represent unlabeled samples of size $m_i$ sampled from each of the two domains respectively. The total number of samples in $X_{train}$ is $N$, with a sample number ratio of $\lambda = (\lambda_0, \lambda_1)$ in each component. If $\hat{h} \in \mathcal{H}$ minimizes the empirical weighted error $\hat{e}_\omega(h)$ using the weight vector $\omega = (\omega_0, \omega_1)$ on $X_{train}$, and $h_j^* = \arg\min_{h \in \mathcal{H}} e_j(h)$ is the optimal hypothesis within the $j$-th domain, then for any $\delta \in (0, 1)$, with a probability exceeding $1 - \delta$, the following holds:*

$$e_j(\hat{h}) - e_j\left(h_j^*\right) \leq \sum_{i=0, i \neq j}^{1} \omega_i(\hat{d}_{\mathcal{H} \triangle \mathcal{H}}\left(S_i, S_j\right) + 4\sqrt{\frac{2d \log(2m) + \log\frac{2}{\delta}}{m}} + \varepsilon_{ij}) + C. \quad (5)$$

$$where\ C = 2\sqrt{\left(\sum_{i=0}^{1} \frac{\omega_i^2}{\lambda_i}\right)\left(\frac{d \log(2N) - \log(\delta)}{2N}\right)}\ and$$

$$\varepsilon_{ij} = \min_{h \in \mathcal{H}} \left\{e_i(h) + e_j(h)\right\}$$

**Remark.** The domain error is determined by three factor: the distribution of training data ($C$), estimated distribution shift ($\hat{d}_{\mathcal{H} \triangle \mathcal{H}}\left(S_i, S_j\right)$) and the performance of the joint hypothesis ($\varepsilon_{ij}$). The ideal joint hypothesis error $\varepsilon_{ij}$ assesses the intrinsic adaptability between domains. Additional theoretical analysis can be found in Appx. G.

Furthermore, Thm. 1 can be used to derive bounds for the test domain error $e_T$. When considering the optimal test hypothesis $h_T^* = \arg\min_{h \in \mathcal{H}} e_T(h)$, we obtain

$$\left|e_T(\hat{h}) - e_T\left(h_T^*\right)\right| \leq \omega_0\left(\hat{d}_{\mathcal{H} \triangle \mathcal{H}}\left(S_0, S_T\right) + 4\sqrt{\frac{2d \log(2m) + \log\frac{2}{\delta}}{m}} + \varepsilon\right)$$

$$+ 2\sqrt{\frac{\omega_0^2}{\lambda_0} + \frac{(1 - \omega_0)^2}{1 - \lambda_0}}\sqrt{\frac{d \log(2N) - \log(\delta)}{2N}}. \quad (6)$$

Thm. 1 formally defines the domain error $e_j(\hat{h})$, and furthermore, we can utilize the test domain error $e_T(\hat{h})$ to verify the significance of incorporating labeled data. The following theorem presents a direct theoretical guarantee that LLMTTT decreases the error bound on the test domain compared to traditional TTT in the absence of labeled test data.

**Theorem 2.** *Let $\mathcal{H}$ be a hypothesis class with a VC-dimension of $d$. Considering the LLMTTT data domains $X_s$ and $X_t$, if $\hat{h} \in \mathcal{H}$ minimizes the empirical weighted error $\hat{e}_\omega(h)$ using the weight vector $\omega$ on training set $X_{tr}$, let the $\epsilon(\omega, \lambda, N)$ to denote the upper bound of $\left|e(\hat{h}) - e\left(h^*\right)\right|$. In the FTTT scenario, no samples from the test domain are selected for labeling (i.e., for weight and sample ratio vectors $\omega'$ and $\lambda'$, $\omega'_0 = \lambda'_0 = 1$ and $\omega'_1 = \lambda'_1 = 0$). Then in LLMTTT , for any $\lambda \neq \lambda'$, there exist a weight vector $\omega$ shuch that:*

$$\epsilon_T(\omega, \lambda, N) < \epsilon_T\left(\omega', \lambda', N\right). \quad (7)$$

**Remark.** Thm. 2 suggests that even a small number of labeled examples during the testing phase can improve the overall model performance, thereby validating the effectiveness of the proposed LLMTTT in addressing distribution shifts. All proofs are provided in Appx. G.

## 5 EXPERIMENT

This section presents extensive experiments to evaluate the performance of the proposed LLMTTT . Firstly, we provide a detailed explanation of the experimental setup. Then, the investigation aims to answer the following research questions:

**RQ1**. How effective is LLMTTT on OOD generalization scenario?
**RQ2**. How to design prompts to obtain high-quality pseudo labels?
**RQ3**. How does the node set used for training affect LLMTTT performance?
**RQ4**. What are the contributions of the two-stage training strategy in LLMTTT framework?

### 5.1 Experimental Settings

*5.1.1 Datasets.* We adopt the following TAGs datasets for node classification: CORA [34], PUBMED [40], CITESEER [10], WIKICS [35] and OGBN-ARXIV [22]. Inspired by GOOD [13], we explicitly make distinctions between covariate and concept shifts and design data splits that accurately reflect different shifts. We used two domain selection strategies combined with covariate and concept, then obtain 4 different splits. For specific details regarding the OOD datasets, please refer to Appendix A. Subsequently, we present the results for concept_degree and covariate_word in Table 1. Additional experimental results can be found in Table 9 in Appendix E.

*5.1.2 Evaluation and Implementation.* We adopt the wide-used metric, i.e., accuracy (ACC) to evaluate the model performance. All experiments were conducted five times using different seeds and the mean performance is reported. GPT-3.5-turbo-0613 is adopted to generate annotations. Regarding the prompting strategy for generating annotations, the integration of cost and accuracy led to the adoption of the few-shot strategy. The budget for active selection was detailed in Table 4 in Appendix A. The pipeline can be applied to any GNN model, with the most popular GCN [27] being adopted in this experiment. The results of other GNN backbones (GAT and

**Table 1: The comparison results between LLMTTT and representative baselines.**

| | concept_degree | | | | | covariate_word | | | | |
|---|---|---|---|---|---|---|---|---|---|---|
| dataset | LLMTTT | EERM | Gtrans | Tent | HomoTTT | LLMTTT | EERM | Gtrans | Tent | HomoTTT |
| cora | **88.53±0.01** | 88.44±0.98 | 85.75±0.02 | 87.21±0.00 | 87.04±0.00 | **92.25±0.02** | 92.14±0.40 | 90.04±0.11 | 90.41±0.00 | 90.51±0.00 |
| pubmed | **86.22±0.00** | OOM | 79.64±0.13 | 85.09±0.00 | 85.09±0.00 | **86.97±0.01** | OOM | 79.44±0.11 | 86.56±0.00 | 86.49±0.00 |
| citeseer | **79.67±0.00** | 69.30±1.81 | 69.43±0.23 | 70.48±0.00 | 70.48±0.00 | **86.33±0.10** | 71.94±0.78 | 69.43±0.23 | 75.86±0.00 | 76.02±0.00 |
| wikics | **80.02±0.02** | 79.89±0.10 | 75.68±0.23 | 78.63±0.00 | 78.89±0.00 | **86.35±0.00** | 85.44±0.23 | 79.77±0.10 | 82.27±0.00 | 82.45±0.00 |
| ogbn-arxiv | **73.82±0.00** | OOM | 63.81±0.21 | 65.40±0.00 | 66.74±0.00 | **75.06±0.00** | OOM | 69.98±0.12 | 70.16±0.00 | 70.32±0.00 |

Graph SAGE) are detailed in Appendix E. Instead of undergoing complex parameter tuning, we fixed the learning rate used in prior studies for all datasets. The code and more implementation details are available in supplementary material.

*5.1.3 Baselines.* We compare LLMTTT with baseline approaches, including (1) EERM [56], a recent State-Of-The-Art (SOTA) method specifically designed for graph OOD issues. (2) Tent [49], a test-time training method in the field of image classification. (3) GTrans [25], a test-time graph transformation approach for node classification. (4) HomoTTT [62], a fully test-time training method that utilizes Self-Supervised Learning (SSL) to fine-tune pretrained GNN model.

## 5.2 Performance on OOD Generalization (RQ1)

To answer RQ1, we conduct a comparative analysis with other four OOD generalization methods. The ACC results are reported in Table 1. From the comparison results, we make some findings. (1) The results in Table 1 display that the proposed LLMTTT performs exceptionally well in the node classification task on the OOD datasets, surpassing all baseline methods. (2) EERM exhibits good performance compared with Tent, but it is constrained by computing resources. This further suggests that post-hoc methods (i.e. FTTT) are better suited for real-world applications due to their plug-and-play simplicity, which does not interfere with the costly training process associated with pre-trained models. (3) GTrans and HomoTTT are both FTTT-based methods. The superior performance of LLMTTT over them illustrates that even a limited number of labeled test instances can significantly enhance test-time training performance.

More results under other split methods are presented in Table 9 in Appendix E. This further demonstrates the effectiveness of the proposed LLMTTT .

## 5.3 Performance on Different Prompts (RQ2)

It is intuitively believed that higher quality pseudo labels can more effectively assist in model fine-tuning during TTT. However, LLMs cannot always produce high-quality labels. Therefore, it is necessary to subsequently obtain better quality labels comparing the accuracy of labels generated by LLMs under different prompts. Before proceeding, we can evaluate this conjecture with a simple experiment.

*5.3.1 The Importance of Pseudo Label Accuracy.* At this part, the relationship between LLMTTT performance and LLM accuracy is explored. After securing a fixed node candidate set, the accuracy of the selected nodes is artificially controlled. The experimental

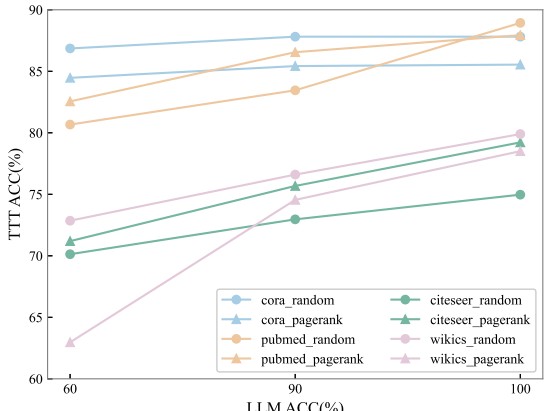

**Figure 2: Investigation on how different LLM accuracy affect the performance of LLMTTT . "random" means the random-based selection. "pagerank" means the pagerank-based selection.**

results in Fig. 2 confirm our conjecture, which states that pseudo label accuracy decide the celling of LLMTTT accuracy under a fixed node selection method

*5.3.2 LLM and TTT Accuracy under Different Prompts.* In this part, we test the accuracy of pseudo labels provided by LLM under different prompts on test samples. Specifically, we used the following prompts: (1) zero-shot; (2) few-shot; (3) few-shot with GNN; (4) few-shot with 2-hop summary. Briefly speaking, "zero-shot" denotes there is no ground-truth label information, while "few-shot" represents that there are some ground-truth labels from the training set. In addition, "few-shot with GNN" further incorporate the information from the pre-trained GNN model based on "few-shot". "few-shot with 2-hop summary" refers to a twice-request prompt strategy [6], which include both the target node information and the aggregation information from its neighboring nodes.

We conducted a comparative study to identify strategies that are effective in terms of both accuracy and cost-effectiveness.

**Observation 1.** The benefits of neighborhood summary are not universal across all datasets. The results presented in Table 2 demonstrate that using a few-shot prompt to aggregate neighbor information can result in performance improvements. However, prompts

**Table 2: Accuracy of pseudo labels annotated by LLMs under different prompts. (·) indicates the cost, which is determined by comparing the token consumption to that of zero-shot prompts.**

| dataset | zero-shot | few-shot | few-shot with GNN | few-shot with 2-hop summary |
|---------|-----------|----------|-------------------|------------------------------|
| cora | 64.40 (1.0) | 67.03 (2.2) | **86.02** (2.4) | 68.10 (3.1) |
| pubmed | 87.84 (1.0) | **91.23**(2.0) | 75.50 (2.2) | 81.35 (3.2) |
| citeseer | 60.92 (1.0) | 74.03 (2.1) | 65.41 (2.3) | **77.43** (3.3) |
| wikics | 66.02 (1.0) | 65.15 (2.6) | **69.88** (2.7) | 55.05 (3.2) |

**Table 3: The results of different active selection strategies.**

| | our | component | | | AL methods | | |
|---------|-----|-----------|---------|--------|------------|---------|--------|
| | hybrid | pagerank | featprop | entropy | random | density | degree |
| cora | **87.34** | 86.62 | 86.86 | 87.10 | 86.62 | 86.62 | 86.86 |
| pubmed | 82.52 | 81.22 | 81.32 | 83.32 | **85.94** | 83.56 | 81.27 |
| citeseer | **76.85** | 69.07 | 75.21 | 76.39 | 73.08 | 72.37 | 69.42 |
| wikics | **74.15** | 73.22 | 72.73 | 73.70 | 72.76 | 74.10 | 73.22 |

incorporating structural information may also be adversely affected by heterogeneous neighboring nodes, as evident by the significant degradation of LLM's performance on PUBMED and WIKICS after incorporating structural information.

The integrated prompt, which combines the predictive information from a pre-trained GNN model, does not consistently yield positive results across all datasets. Additionally, its performance in this scenario is intricately tied to the effectiveness of the pre-trained model.

Given the aforementioned information and the cost under different prompts shown in Table 2, we adopt the few-shot prompt approach with the aim of attaining a more generalized and superior performance. Meanwhile, the failure of "few-shot with 2-hop summary" also motivated us to design a prompt that can accurately represent the graph structure. Thus, LLMs can be more effectively employed to solve graph level tasks.

## 5.4 Impact of Candidate Node Set (RQ3)

The nodes utilized for model training undergo two selection processes. Initially, a hybrid active node selection strategy is employed, followed by a post-filtering strategy that leverages the prediction results obtained from LLMs.

*5.4.1 Impact of Active Selection Strategies.* Initially, we explore various components of hybrid active node selection, including Pagerank, FeatProp, and entropy. Secondly, we compared our hybrid node selection strategy with traditional active learning methods, such as density and degree, as well as random node selection methods.

From Table 3, we find that traditional active learning methods are not applicable and effective as expected in our scenarios. Based on the empirical results, the study makes the following observation:
**Observation 2.** Some research [7] has demonstrated that nodes in proximity to cluster centers often demonstrate higher annotation

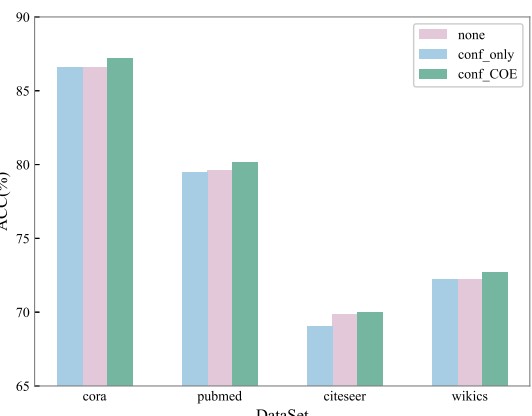

**Figure 3: The results of different post-filtering strategies. "none" means graph active selection combined without post-filtering. "conf_only" means the graph active selection combined with confidence. "conf_COE" means the graph active selection combined with confidence and COE.**

quality. Consequently, the node set selected by density-based active strategy will be assigned high-quality annotations. However, the density-based active selection strategy does not achieve the optimal performance. This gives us an intuition that improvement not only depends on LLM accuracy, but also the node selection. The Appendix C further substantiates our conjecture through the control of accuracy of labels annotated by LLMs.

*5.4.2 Impact of Post-Filtering.* In this part, we examine the effectiveness of the proposed post-filtering strategy. Given that the proposed post-filtering strategy incorporates confidence scores and takes into account the diversity of nodes (COE), we also perform ablation experiments in this section. The experimental results are presented in Figure 3.
**Observation 3.** The proposed post-filtering strategy demonstrates significant effectiveness. Furthermore, aligning with our previous observation, although the node selected by "conf_COE" does not possess the most accurate labels, they demonstrate the best model performance. This verification from another perspective suggests that model performance is not fully positively correlated with pseudo label accuracy.

## 5.5 Ablation of Two-stage Training (RQ4)

Our method considers a two-stage training strategy for model adaptation including training with filtered nodes and self-training with unlabeled nodes. To verify the effectiveness of each stage, we perform an ablation study to investigate whether incorporating training with filtered nodes or self-training strategy can lead to performance improvements. The results in Figure 4 indicate that both of the training strategy contribute to the model performance, with stage 1 making a greater contribution. This not only underscores the effectiveness of our proposed two-stage training strategy but also further highlights that the incorporating limited labeled test instances enhance model performance.

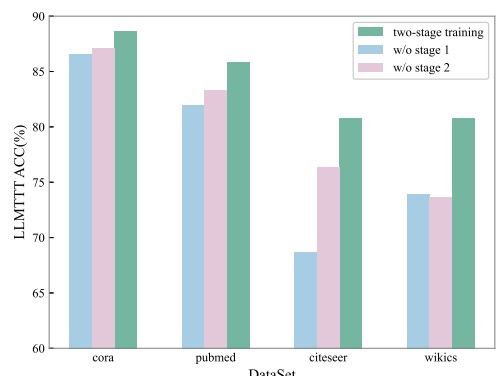

**Figure 4: Effectiveness of two-stage training**

# 6 RELATED WORK

LLMTTT aims at solving the challenges of data distribution shift in GNNs via a novel test-time training method based on LLMs. To achieve this goal, a careful graph active learning strategy is also developed. The related work are discussed as follows:

## 6.1 Distribution shift in GNNs

Graph Neural Networks (GNNs) have demonstrated exceptional capabilities in graph representation learning [36, 59], achieved revolutionary progress in various graph-related tasks [51], such as social network analysis [24, 45], recommendation systems [9, 18, 53], and natural language processing [3, 23, 29]. However, a distribution shift has been observed in various graph-related applications [20, 21], where the graph distribution in the training set differs from that in the test set. Such discrepancy could substantially degrade the performance of both node level [56, 66] and graph level tasks [56, 66]. This distribution shift frequently occurs between the testing and training graphs [20, 21]. Therefore, enhancing the out-of-distribution (OOD) generalization capabilities of GNNs is crucial. Several solutions have been proposed to tackle this issue, such as EERM [56], which trains GNNs to be adaptable to multiple environments by introducing environmental variables, and GTrans [25], which enhances generalization ability by modifying the input feature matrix and adjacency matrix during test time.

## 6.2 Test-Time Training

Test-time training (TTT) is a technique recently proposed for partially adapting a model based on test samples, to account for distribution shifts between the training and test sets. TTT was first introduced by [44]. To address the unexpected adaptation failures in TTT, TTT++[31] employs offline feature extraction and online feature alignment to enable regularization adaptation without the need to revisit the training data. However, in some cases, the training data may be unavailable during test time or the training process may be computationally demanding, which can reduce the applicability of these methods. To overcome this limitation, Tent [49] introduces a method for fully test-time training that relies solely on test samples and a trained model. They propose an online setting after the TTT task to achieve fully test-time training through the minimization of the model's test entropy. While the aforementioned

studies focus on test-time training within the image domain, the TTT framework has also been implemented in the realm of graphs, including GTrans [25], GT3 [52], GraphTTA [4], and TeSLA [46].

## 6.3 Graph Active Learning

Graph active learning aims to optimize test performance through the strategic selection of nodes within a constrained query budget, effectively addressing the challenges of data labeling. The most prevalent approach in active learning is uncertainty sampling [37, 43, 48, 54], wherein nodes that the current model has the least certainty about are selected during the training phase. Another significant strand within active learning approaches involves distribution-based selection strategies. These methods [2, 11, 41, 42, 65, 65] evaluate samples based on their positioning within the feature distribution of the data. Representativeness and diversity represent two commonly utilized selection criteria, both of which rely on the data distribution. Generally, active learning primarily focuses on selecting representative nodes; however, it faces additional challenges in real world scenarios. Furthermore, active learning needs to address two key issues: assigning pseudo-labels to the selected nodes and effectively utilizing a limited number of labels for training. In the proposed LLMTTT , these two problems are well solved.

## 6.4 LLMs for Graphs

Large language models (LLMs) with massive knowledge demonstrate impressive zero-shot and few-shot capabilities. Considerable research [14, 17] has begun to apply LLMs to graphs, enhancing performance on graph-related tasks. Utilizing LLMs as enhancers [14] presents a viable approach, leveraging their power to enhance the performance of smaller models more efficiently. Compared to shallow embeddings, LLMs offer a richer commonsense knowledge base that could potentially enhance the performance of downstream tasks. Relying solely on LLMs as predictors [6, 50, 60] represents another viable approach, with GPT4Graph [14] evaluating the potential of LLMs in performing knowledge graph (KG) inference and node classification tasks. NLGraph [50] introduced a comprehensive benchmark to assess graph structure reasoning capabilities. Distinct from these approaches, we employ LLMs as annotators as [7], combining the advantages of the two aforementioned methods to train an efficient model without relying on any true labels.

# 7 CONCLUSION

We introduce a novel TTT pipeline LLMTTT which introduces LLM as an annotator to provide a limited number of pseudo labels for fine-tuning the pre-trained model. To select a candidate set that is both representative and diverse, the proposed pipeline LLMTTT designs a hybrid active selection that also considers the pre-trained model signal. Following this, we generate high-quality labels with corresponding confidence scores with the help of LLMs. Finally, we present a two-stage training strategy that maximises the use of the test data. The strategy includes confidence-based post-filtering to mitigate the potential impact from the noisy labeled test data. Additionally, a weighting function is used to introduce a large amount of unlabeled test data into the training process. Comprehensive experiments and theoretical analysis demonstrate the effectiveness of LLMTTT .

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
