# OpenReview forum: "Test-Time Training on Graphs with Large Language Models (LLMs)"
_acmmm.org/ACMMM/2024/Conference — MM2024 Poster_

### Official Review · Reviewer_Q3AW · 2024-05-24

**Rating:** 4
**Confidence:** 3

**Summary:**

This paper proposes novel methods for testing out-of-distribution generalization. It introduces LLMTTT (Large Language Models Test-Time Training), a method aimed at improving the effectiveness of Graph Neural Networks (GNNs) in handling distribution shifts between training and test data. LLMTTT leverages the annotation capabilities of Large Language Models (LLMs) on Text-Attributed Graphs for test-time adaptation.     It introduces a hybrid active node selection strategy and a two-stage training strategy to tailor the test-time model using limited and noisy labels obtained from LLM annotations. The method is supported by theoretical analysis and extensive experiments demonstrating significant performance improvements compared to existing approaches.

**Strengths:**

S1. The paper introduces a novel method, LLMTTT, which addresses the challenge of distribution shifts in GNNs by leveraging LLMs for test-time adaptation. This approach is innovative and demonstrates a new direction in improving GNN generalization performance.  By leveraging the annotation capabilities of LLMs on TAG, LLMTTT taps into the rich contextual understanding provided by these models, enhancing the adaptability of GNNs.

S2. LMT is supported by extensive experimental results demonstrating significant performance improvements compared to existing approaches. This empirical validation strengthens the credibility and applicability of the proposed method.

S3. The paper provides theoretical analysis to support the validity of the proposed method, enhancing the understanding of its underlying principles and mechanisms.

**Limitations:**

W1. The paper does not provide a detailed analysis of the computational overhead and scalability implications of LLMTTT resulting from its reliance on LLMs.

W2. The two-stage training strategy of LLMTTT relies on annotations from LLMs, which may introduce noise or inaccuracies, especially in cases where LLMs struggle with understanding complex graph structures or context.  Noisy annotations could negatively impact the performance of LLMTTT, particularly in scenarios with limited labeled data.

W3. There are some grammar errors, such as, "Appendix" is used in the first part of the paper, and "Appx" is used in the second part of the paper. Please pay attention to the consistency of the statement before and after the paper, which will greatly improve the readability of the paper.

W4. The notation in Algorithm 1 is not stated. Providing more notation for formulas in the paper will improve the clarity of the paper.

W5. The code is not provided in the supplementary material, which affects the reproducibility of the paper.

**Suitability:**

3

---

### Official Review · Reviewer_bvpF · 2024-05-24

**Rating:** 4
**Confidence:** 3

**Summary:**

This paper presents LLMTTT, which leverages LLM as an annotator to improve a set of pseudo labels for model fine-tuning. To curate a candidate set that embodies both representation and diversity, LLMTTT employs a hybrid active selection method that takes into account the signal from the pre-trained model. Subsequently, LLM produces higher quality labels. And a two-stage training approach is adopted for optimizing test utilization.

**Strengths:**

Test time training on graph is an excellent entry point; even as research on TTT has progressed, there are very few studies on improving TTT with LLM. I believe out-of-distribution generalization is a highly significant topic.

The extensive experiments validate the effectiveness of the proposed method over other SOTA methods.

**Limitations:**

1. I am not sure about the correctness of proof for Theorem 1 since I cannot find the Appx from the submission. I still have some confusion here, for example, the how you estimated distribution shift in theorem 1 and the derivation of boundness of this term.


2. Why the nodes with higher prediction entropy are useful for test-time training? If the test-time input graph/node is out-of-distribution, are you assuming that some information from in-distribution graph could still help? What if this assumption does not hold anymore?

3. Is the LLM here really an effective design? It is better to prove this with a simple experiment.

**Suitability:**

3

---

### Official Review · Reviewer_kyxj · 2024-05-24

**Rating:** 3
**Confidence:** 3

**Summary:**

The paper introduces a novel method called LLMTTT (Test-Time Training on Graphs with Large Language Models) to address the challenge of Out-of-Distribution (OOD) generalization in Graph Neural Networks (GNNs). LLMTTT leverages Large Language Models (LLMs) to provide pseudo-labels for a carefully selected set of nodes during the test-time adaptation phase, avoiding the need for intricate unsupervised task designing. The paper proposes a hybrid active node selection strategy that considers node diversity, representativeness, and prediction signals from the pre-trained model, and a two-stage training strategy to utilize both the noisy labeled nodes and the remaining unlabeled nodes during test-time adaptation. Extensive experiments and theoretical analysis demonstrate the effectiveness of LLMTTT in improving OOD generalization on various graph datasets.

**Strengths:**

Strengths:

- The paper introduces a novel method called LLMTTT that leverages LLMs to provide pseudo-labels for a carefully selected set of nodes during the test-time adaptation phase, avoiding the need for intricate unsupervised task designing. This approach allows the model to benefit from the annotation capabilities of LLMs to enhance test-time training and handle the Out-of-Distribution (OOD) problem on graphs.
- The paper proposes a hybrid active node selection strategy that considers node diversity, representativeness, and prediction signals from the pre-trained model. This strategy helps to select the most valuable nodes for annotation using LLMs within a constrained budget.
- The paper provides theoretical analysis that demonstrates the effectiveness of LLMTTT in improving OOD generalization on various graph datasets. Specifically, Theorem 2 presents a direct theoretical guarantee that LLMTTT decreases the error bound on the test domain compared to traditional Test-Time Training (TTT) in the absence of labeled test data.

**Limitations:**

Weakness:
- The author does not provide the source codes of LLMTTT, so the experimental results can not be reproduced. Additionally, the baselines are not strong enough, and the number of baselines is limited. More effective, latest, and novel GNN methods for OOD need to be considered, such as OOD-GNN[1], OODGAT-ATT[2], and OOD-GMixup[3].
- Compared with EERM, the experimental results from LLMTTT in Cora and Wikics are not obviously enhanced compared to those in Citeseer in Table 1, and the author does not provide a detailed explanation for this phenomenon.
- The author only evaluates LLMTTT on a limited set of graph datasets, such as Cora, PubMed, Citeseer, and WikiCS. While these are commonly used benchmarks, evaluating the method on a more diverse range of graph datasets, including larger and more complex graphs, would provide a more comprehensive understanding of its generalization capabilities.
- The paper also does not provide the ablation studies about the key components of the LLMTTT framework, such as the hybrid active node selection strategy and the two-stage training approach.

References:

- [1] Li H, Wang X, Zhang Z, et al. Ood-gnn: Out-of-distribution generalized graph neural network[J]. IEEE Transactions on Knowledge and Data Engineering, 2022.
- [2] Song Y, Wang D. Learning on graphs with out-of-distribution nodes[C]//Proceedings of the 28th ACM SIGKDD Conference on Knowledge Discovery and Data Mining. 2022: 1635-1645.
- [3] Lu B, Zhao Z, Gan X, et al. Graph out-of-distribution generalization with controllable data augmentation[J]. IEEE Transactions on Knowledge and Data Engineering, 2024.

**Suitability:**

3

---

### Official Review · Reviewer_T5Zm · 2024-05-25

**Rating:** 2
**Confidence:** 3

**Summary:**

This paper proposes a pipeline LLMTTT, which incorporates LLM as an annotator to generate  pseudo labels for test-time training. LLMTTT incorporates a hybrid active selection method and generate high-quality labels along with confidence scores. Additionally, a two-stage training strategy is introduced to maximize the use of the test data.

**Strengths:**

- This paper is well-written and it provides a detailed experimental analysis.
- Theoretical analyses are provided to support the effectiveness of labelled test samples
- The proposed LLMTTT is a novel pipeline that offers valuable insights into LLM for graph OOD.

**Limitations:**

- In **The Selection Algorithm**, the authors don’t specify whether the PageRank and FeatProp scores are calculated over all samples or $\beta B$ samples. If the scores are calculated within the $\beta B$ samples, it may not guarantee the diversity of the resulting $B$ samples.
- There are some errors in the paper, “incorporates both the labeled test nodes $D_{tr}$ and unlabeled test nodes **$D_{\text {tre}}$**”, “This section theoretically demonstrates that incorporating labelled test samples provided by **LMMs** during the test-time training phase can significantly improve the overall performance across the test domain.”
- The topic may be not relevant to the MM conference since the tasks in this paper do not include multi-modal information.
- The inference time in TTT could be a concern.
- LLM may have trained on some data that are adopted for testing, e.g., arxiv. Therefore, it could be unfair to compare the OOD performance for this method (w/ llms) and baselines (w/o llms).

**Suitability:**

1

---

### Meta-Review · Area_Chair_EL8x · 2024-07-09

**Recommendation:** Accept (Poster)
**Confidence:** 3

**Metareview:**

The paper proposes LLMTTT, a novel pipeline for test-time training of Graph Neural Networks (GNNs) using Large Language Models (LLMs) to generate pseudo-labels for out-of-distribution (OOD) generalization. Reviewers appreciated the detailed experimental analysis, theoretical support, and innovative use of LLMs for enhancing GNN adaptability. However, concerns were raised regarding the lack of experiments on multimodal datasets, potential biases from LLM training data, limited baseline comparisons, and the absence of ablation studies. Additionally, the computational overhead and scalability of LLMTTT were not thoroughly analyzed. The consensus leaned towards borderline acceptance, acknowledging the method's potential despite some limitations in experimental scope and relevance to multimodal information processing.

There is a significant debate about this paper's relevance to the MM community. While graph is not directly an MM topic, it surely connects important research areas such as multi-modal data application. Thus, we should not solely judge the paper based on the relevance. The final decision is borderline acceptance.